# Oleaginous Heterotrophic Dinoflagellates—Crypthecodiniaceae

**DOI:** 10.3390/md21030162

**Published:** 2023-02-28

**Authors:** Alvin Chun Man Kwok, Stanley Ping Chuen Law, Joseph Tin Yum Wong

**Affiliations:** 1Division of Life Science, The Hong Kong University of Science and Technology, Clear Water Bay, Kowloon, Hong Kong; 2State Key Laboratory of Molecular Neuroscience, The Hong Kong University of Science and Technology, Clear Water Bay, Kowloon, Hong Kong

**Keywords:** *Crypthecodinium*, Crypthecodiniaceae, croucheri, DHA, oleaginous

## Abstract

The heterotrophic *Crypthecodinium cohnii* is a major model for dinoflagellate cell biology, and a major industrial producer of docosahexaenoic acid, a key nutraceutical and added pharmaceutical compound. Despite these factors, the family Crypthecodiniaceae is not fully described, which is partly attributable to their degenerative thecal plates, as well as the lack of ribotype-referred morphological description in many taxons. We report here significant genetic distances and phylogenetic cladding that support inter-specific variations within the Crypthecodiniaceae. We describe *Crypthecodinium croucheri* sp. nov. Kwok, Law and Wong, that have different genome sizes, ribotypes, and amplification fragment length polymorphism profiles when compared to the *C. cohnii*. The interspecific ribotypes were supported by distinctive truncation-insertion at the ITS regions that were conserved at intraspecific level. The long genetic distances between Crypthecodiniaceae and other dinoflagellate orders support the separation of the group, which includes related taxons with high oil content and degenerative thecal plates, to be ratified to the order level. The current study provides the basis for future specific demarcation-differentiation, which is an important facet in food safety, biosecurity, sustainable agriculture feeds, and biotechnology licensing of new oleaginous models.

## 1. Introduction

*Crypthecodinium cohnii* sensu lato is employed for the industrial production of omega-3 docosahexaenoic acid (DHA), which is crucial for neuronal and infant health [1]. *C. cohnii* can accumulate lipids with high DHA content [2,3,4,5] and was the first algae approved by U.S. Food and Drug Administration for DHA production. Dinoflagellates from the family Crypthecodiniaceae, hitherto with only one partially described species, *Crypthecodinium cohnii* (=*Crypthecodinium setense*) [6], is composed of multiple undescribed taxonomic units (OTUs) commonly isolated from decaying seaweed on shore [7,8,9,10,11]. Although it is widely recognized that the family contains species with differences in their SSU rDNA sequences [12], many Crypthecodiniaceae taxons were ribotyped solely with either the conserved LSU rDNA or SSU rDNA marker, but not the more variable ITS phylogenetic marker for the analysis of intra-species variations. Moreover, intraspecific delineation by thecal plate tabulation is also hampered by the possession of thin thecal plates [9], and often with ill-defined thecal plates. 

Many Crypthecodiniaceae members are found on intertidal seaweed [7,13], qualifying them as the only amphibious dinoflagellate group. There are no reports of toxin production from the family and *C. cohnii* has been used in feeds for aquaculture [14,15]. Carotenoids are lipophilic, colored pigments, which are widely distributed in higher plants and microalgae [16]; they have been used as food additives for nutraceutical and pharmaceutical applications (e.g., for color; additionally, lutein and zeaxanthin are effective for the prevention of eye disease, and fucoxanthin exhibits a strong anti-obesity activity), feeds, and as natural colorants in cosmetics industries [17]. As in some microalgae [18,19,20,21], Crypthecodiniaceae cells contain the anti-cancer, provitamin A carotenoids (β-carotene and γ-carotene) [22,23] and was proposed as a production model [22].

Species-level identification is a food safety requirement in relation to PUFA (polyunsaturated fatty acids) sources, as demonstrated by a recent rejection from a regulatory agency [24]. The family was commonly placed within the order Gonyaulacales [25] without formal description; the order contains dinoflagellates with theca showing a latitudinal series of plates [26]. Compatibility studies suggested *Crypthecodinium* spp. likely contained independent species [11], but no further characterization was conducted. This taxonomic position is thus counterintuitive, given its industrial and cell biology significance, with the organisms being the only colony-forming dinoflagellates with published mutants [27,28,29,30,31,32,33], and given the availability of the inducible strain crossing method [33], a true minimal medium without the use of undefined seawater [34], the cell cycle synchronization method, as well as efficient transformation protocols [29,35,36,37,38]. Relatively high oil content is an apparent genus trait, as all the strains in our laboratory collection exhibited at least one order of magnitude higher Nile red intensity than other dinoflagellates, indicating the group’s potential for biofuel production. Stringent rectification of independent taxons would not only allow future biotechnology exploitation, but also safeguard the deployment of existing species-specific technology. 

We report here phylogenetic analysis of Crypthecodiniaceae based on three ribosomal markers [39,40]. Cumulated with morphological characters, thecal plate formulas, AFLP and three molecular markers, as well as genome size differences, we report *Crypthecodinium croucheri* sp. nov. Kwok, Law, and Wong, as well as propose two new species from the Rykuyu.

## 2. Results

Exposed low-tide epiphytes collected from local shores were briefly washed with seawater prior to isolation of dinoflagellate cells. Isolated cells were incubated without light. The present paper focused on those isolates with little pigment and a crypthecodinoid shape, which were further purified with dilution passages in a 96-well plate without light. Earlier observations suggested differences in size ranges (Table 1) with the control *C. cohnii* strains, inspiring us to conduct ribotype surveys. In the following description, we describe *Crypthecodinium croucheri* sp. nov. Kwok, Law, and Wong and proposed *Crypthecodinium okinawan* sensu lato (will be referred to as *C. okinawan*) and *Crypthecodinium ryukyu* sensu lato (will be referred to as *C. rykuku*) (see later justifications).

### 2.1. Genetic Distances Suggest Interspecific Demarcations within Crypthecodiniaceae

Despite its importance in the certification for industrial licensing, there has been relatively little study of inter- and intra-specific variations between different OTUs of the genus. Ribotypes are well recognized to trace dinoflagellate evolution, with ITS region (ITS1-5.8S-ITS2) successfully tested to demarcate species between morphological similar taxons [40,41,42,43].

Previous genetic distance analyses of dinoflagellates ITS amongst 14 genera suggested that ITS region p-distances ≥ 0.04 (substitutions per site) are indicative of species-level divergences [42]. The p-distance values between C. cohnii and C. croucheri (0.131 to 0.150), *C. okinawan* (Rykuyu Crypthecodinium sp. Group II [12]) and *C. cohnii* (0.238–0.304), *C. rykuyu* (Rykuyu Crypthecodinium sp. Group I [12]) and *C. cohnii* (0.054–0.068), *C. okinawan* and *C. croucheri* (0.253–0.314), *C. rykuyu* and *C. croucheri* (0.178–0.198), Crypthecodinales R6 and the rest of Crypthecodiniaceae (0.997–1.193) (Figure 1), were much higher than the 0.04 species-level divergences cutoff [42], strongly supporting their inter-specific demarcation. The genetic distance would be further increased if more geographically distant *C. cohnii* were included, when compared to the current Hong Kong *C. cohnii* isolate (e.g., Tolo Harbor). We expect further inter-specific level in other *C. cohnii* sensu lato taxons.

Intraspecific p-distance values for the three C. cohnii strains (CC1, ATCC30021 and ATCC30556) ranged from 0 to 0.023, whereas p-distance values between the four *C. croucheri* strains ranged from 0.007 (HKUST-1002, 1003 and 1006) to 0.063 (HKUST-1001) (Figure 1). High ITS region p-distance values were also observed within Crypthecodiniaceae but we felt that different strains of the same species, if they were available (as for *C. okinawan, C. rykuyu* and *C. croucheri*) would be best fitted to substantiate stabilization of the highly-variable ITS regions. Another likely undescribed species is marked by the higher p-distance value of the *C. cohnii* CCMP316 (>0.278 when comparing to all other Crypthecodiniaceae members (Figure 1)).

It must be noted that Group II Rykuyu isolates [12] apparently had extra ITS changes from the Group I Rykuyu isolates, *C. croucheri* and *C. cohnii*, in addition to sequence differences at the blue-boxed region (Figure 2). This, and the three ribotypical demarcations suggest that the Rykuyu isolates Group I and II should be independent, and we propose *Crypthecodinium ryukyu* [12] sensu lato for Group I, and *Crypthecodinium okinawan* [12] sensu lato for Group II, awaiting formal description from the original isolators, as the demarcations have been recognized previously [12]. The presence of substantial sequence truncation and nucleotide insertions in the two internal transcribed spacers ITS1 and ITS2 in all the strains suggested stable taxons, whether or not the truncations-deletion were of recent events. The truncation-insertion events (Figure 2) would not be fully reflected on a phylogenetic tree with fixed alignment gaps allowances of most parsimonious algorithms. ITS processing is involved in the maturation of 60S pre-ribosomes prior to nuclear export [44], which will be crucial in dinoflagellates, as there is no nuclear envelope breakdown. The changes in ITS1 also co-occurred with changes at ITS2, strongly indicating co-evolved functional significance.

### 2.2. Phylogenetic Analyses Suggested Multiple Clades within Crypthecodiniaceae

We constructed phylogenetic trees with selected taxons that have either one of the three rDNA markers. In phylogenetic analysis, it is apparent that the *Crypthecodinium* taxons contain well defined clades with inter- and intra-species level variations. All three *C. croucheri* ribotypes form a clade (Figure 3A–C) with phylogenetic positions closer to selected Rykuyu *Crypthecodinium* sp. ribotypes than *C. cohnii*. 

When all phylogenetic trees were rooted with *Amphidinium carterae*, a recognized early lineage, the immediate node joining the Crypthecodiniaceae to the main dinoflagellate lineages did not give significant bootstrap values (Figure 3A–C), suggesting the Crypthecodiniaceae long branches are indicative of substantial ITS divergence from the previous ancestral group, and substantiate a potential independent order, of which we propose Crypthecodinales (see the following paragraph). The inclusion of the undescribed R6 ribotypes (a Crypthecodiniaceae from our recent collection) increased the bootstrap values (to 100) of this group (including CAAE-CL2 and *Gyrodinium lebouriae* sensu lato), supporting the differential association between R6 group and the Crypthecodiniaceae. 

Both LSU and SSU rDNA trees also indicated this early branch with total bootstrap value, amid with partially described taxon CAAE-CL2 and *Gyrodinium lebouriae* sensu lato (EF681914.2), respectively (Figure 3B,C). *G. lebouriae* sensu lato also adopts eleutheroschisis cell division as the main *C. croucheri-C. cohnii* group [45], and it did not clade with other *Gyrodinium* spp. (LSUrDNA tree), suggesting prior casual placement was erroneous. Some isolates of *G. lebouriae* sensu lato were phagocytic (forming peduncle and feeding on the prey chloroplasts) [45], as in the case of the Eliat strain [13] of Crythecodiniaceae. We did not observe peduncles in the main *croucheri-cohnii* groups, and *C. cohnii* (ATCC33506) did not uptake fluorescence bacteria (*E. coli*) when fed in seawater medium (data not shown). It is highly likely that the R6 ribotype group, of which we are currently analyzing new members, represents an early branch of the ancestral group sister to Crypthecodiniaceae. Based on SSUrRNA and LSUrRNA phylogenetic trees, we propose the Crypthecodiniales sensu lato, awaiting further confirmation of the R6 group (Figure 3A,B). All Crypthecodinales (including R6 sensu lato) shared the characteristics of poorly developed thecal plates, a distinguishing characteristic not found in the Gonyaudinales.

### 2.3. Crypthecodinium croucheri: Species Novel Kwok, Law and Wong 2023

#### 2.3.1. General Description

Coccoid-swarmer cells were observed in all isolates described herein. Motile swarmer *C. croucheri* cells appeared superficially crypthecodinoid under light microscopy, with hypocone larger than epicone, thinly thecated, and slightly dorsoventrally compressed (Figure 4A). Fluorescence photomicrographs of calcofluor white (CFW)-stained cells suggested *C. croucheri* HKUST strains having a theca composed of thin but structured plates (Figure 4A). The cingulum shows 1-2 girdle width displacement (see later session), and it did not fully encircle the cell, exhibiting a leftward descending spiral that terminated near the right lateral side after traversing roughly two-thirds of the cell’s circumference (see later session). Non-motile coccoids appeared as Crypthecodiniaceae, showing ovoid in ventral or dorsal view, coccoidal divisions and multiple fissions. At least six clonal strains, which were subsequently identified as belonging to *C. croucheri*, were maintained using 1:1 (*v/v*) f/2:rich medium.

*C. croucheri* cells, which did not propagate well in the *C. cohnii* minimal medium MLH [34], were generally smaller than *C. cohnii* (ATCC30556), with ~10% smaller width and length, and ~30% smaller in volume (Table 1). Unlike other *C. cohnii* strains, all *C. croucheri* strains did not form colonies on the *C. cohnii* defined MLH minimal medium [9] agar. *C. croucheri* were able to form colonies on f/2 agar (prepared from red algae Irish moss [46], Agar No. 1, Oxoid), with apparently more multiple fissions cells when comparing the liquid cultures (Figure 4C). Nutritional demarcations, as well as different optimal thermal ranges (Appendix A), are not only important for DHA biotechnology, but also demonstrated potential seasonal niche changes between *C. cohnii* and *C. croucheri*, despite both species being reported for the same location.

#### 2.3.2. Genome Sizes and Amplification Fragment Length Polymorphism (AFLP)

*C. croucheri* haploid genome content was estimated, with flow cytometry with cell-cycle phased comparisons, to be 5.6 pg per cell, significantly (~7.5–12%) lower than *C. cohnii* (Figure 5A,B). Previous dot blot analysis demonstrated that the *C. cohnii* (*C. cohnii* Biecheler 1938, strain Whd) genome contained a high copy number of Cc18 and Cc20 repeated sequences [47], which we deployed in conjunction with amplification fragment length polymorphism (AFLP) analysis for Crypthecodiniaecae species differentiation. AFLP analysis based on Cc20 repetitive sequence did not generate any clearly resolvable AFLP products in *C. croucheri* when compared to *C. cohnii* (Figure 5D,E), whereas Cc18 primers produced distinct and different banding patterns between *C. croucheri* and *C. cohnii*, and with intraspecific differences between different *C. cohnii* members (i and iv in Figure 5C,E), suggesting previously identified repeat primers being good AFLP markers for Crypthecodiniaecae members.

#### 2.3.3. Lipid Content

Nile red is widely deployed to estimate polar and neutral lipid content/ratio, of which yellow/gold emission (≤580 nm, excited by shorter wavelength) wavelengths favor the detection of highly hydrophobic neutral lipids (triacylglyceride, TAG), whereas red emission (≥590 nm, excited by longer wavelength) wavelengths favor a general fluorescence for polar lipids [51]. Significant correlation between the polar lipid (red) fluorescence with DHA determined by chromatography showed that *C. cohnii* cellular DHA is distributed mainly in polar lipid fractions [52,53,54]. Flow cytometric analysis of Nile-red-stained *C. croucheri* cells suggested significantly higher polar and neutral lipid content per cell, when compared to *C. cohnii* (Figure 6B–E). *C. cohnii* ATCC30556 was chosen for comparison, as the strain was isolated in Hong Kong, and has been used for fed-batch fermentation studies for DHA production [55,56,57,58,59,60,61]. *C. croucheri* G_1_ cells had significant higher mean Nile red intensities than G_2_ cells (Figure 6A–E), unlike *C. cohnii* which had similar levels in both cell cycle phases [53]. The G_1_-G_2_ lipid content differences were widened in *C. croucheri*, especially in HKUST-1001 and HKUST-1002 (Figure 6D,E, marked by “∆”), with G_1_ cells accumulating more polar and neutral lipid than G_2_ cells. This is a non-trivial trait for life-cycle manipulation of oil production. Direct measurement of total lipid content (per cell and per dry weight) using the biochemical sulpho-phospho-vanillin (SPV) method [62,63,64], which will include membrane lipids in addition to the lipid bodies that dominated Nile red staining, suggested significantly higher lipid content (per cell and per dry weight) in *C. croucheri*, especially HKUST-1002, when compared to *C. cohnii* (Figure 6F) strains grown under batch culture. Comparing to over 40% lipid content per dry weight in some *C. cohnii* culture grown in fermenter [65], the apparent lower lipid content per dry weight (~27% in a 7-day old HKUST-1002 batch culture) in *C. croucheri* is likely attributable to glucose/nutrient limitation and pH limitation in batch culture. 

#### 2.3.4. Kofoidian Plate Pattern

Fluorescence microscopy of CFW-stained *C. croucheri* cells showed no distinct apical pore complex Po (Figure 7E,G). Within the apical series, the first apical plate 1′ was asymmetrical, and six sided (para-plate), and contacted the anterior sulcal plate Sa, but not the X plate, which was different (also in shape) from *C. cohnii* and the *Crypthecodinium* sp. CAAE-CL2 (Figure 7A–C,M,N). The apical plates 2′ and 3′ were pentagonal, whereas 4′ was quadrangular (Figure 7E–G). The apical plates 2′ and 4′ were separated by plates 1′ and 3′, as described in the original *C. cohnii* description, rather than joined at a point as in the *Crypthecodinium* sp. CAAE-CL2 [9] (Figure 7E,M,N). Anterior intercalary plates 1a and 2a were small, whereas the pentagonal 3a and hexagonal 4a were much larger. Of the six precingular plates, plate 1″ was triangular, 2″ was quadrangular, and 4″ and 5″ were pentagonal, whereas 3″ and 6″ were hexagonal. The cingulum was composed of six comparably sized plates and there was no transitional plate. The X plate was heptagonal and located at the right hypotheca, adjacent to the end of the cingulum. The sulcus consisted of an anterior sulcal plate (Sa), a left sulcal plate (Ss), a right sulcal plate (Sd), a posterior accessory plate (Spa), and a posterior sulcal plate (Sp) (Figure 7H,N). The plate arrangement of hypotheca was slightly asymmetrical. The hypotheca consisted of five postcingular and two antapical plates (Figure 7J,L,N). The postcingular plates consisted of five plates, plates 1′″, 3′″ and 4′″ were quadrangular, whereas 2′″ and 4′″ were pentagonal. The antapical series was composed of two plates, the antapical plate 1″″ was quadrangular whereas 2″″ was hexagonal.

Kofoid thecal plate tabulation of *C. croucheri* was 4′, 4a, 6″, X, 6c, 5s, 5′″, 2″″, which was more similar to the original description of *C. cohnii* Biecheler than was the other *Crypthecodinium* sp. CAAE-CL2 (4′ 4a, 4″, X, 5 or 6c, ?s, 5′″, 1p, 1″″) (Biecheler, 1938; Fensome et al., 1993b; Parrow et al., 2006) [6,9,66] (Figure 7M,N). The mis-labelled anterior intercalary plate 4a in a previous publication (Parrow et al., 2006) [9] should be precingular plate 6″ (Figure 7M,N). Although this formula was similar to the one reported for a *C. cohnii* strain, a number of deviations from the typical plate pattern were also spotted (Figure 7M,N). The main features differentiating *C. croucheri* from *C. cohnii* include: (i) apical plate 1′ did not share a suture with the X plate; (ii) the precingular plate 1″ was split into plates 1″ and 2″, with suture sharing between 1′ and 2″. There were topological differences at 4′-3′-2′ and 1′, intercalations that were not reflected in Kofoidal’s description (Figure 7), as well as the lower contact plate of 1′ either as 5″ (*C. cohnii* = *C. sentense*), 4a″ (*Crypthecodinium* spp. CAAE-CL2) or 6″ (*C. croucheri*).

Variations in number of plates were also observed: (i) the precingular plate 6″ could be split into two plates, 6″_1 and 6″_2 (Figure 7C versus Figure 7A,D); and (ii) the anterior intercalary plate 3a could split into 3a_1 and 3a_2 (Figure 7K versus Figure 7F). 

Degenerative thecal plates, which likely shrink in most fixatives, are a distinctive trait of Crypthecodiniaceae. Despite the tabulation similarity within Crypthecodiniaceae, which was likely constrained with the degenerative thecal plates, the distinct ribotype cladding, genome size differences, AFLP patterns, and nutritional modes strongly support the new isolates as constituting a well-supported novel species. The thinly deposited thecal plates are also not always amenable for gold surface spud-marking [12], and we have not acquired any plate demarcation with scanning electron micrographs for *C. croucheri* despite several attempts, likely attributable to intercalation that was removed upon sample fixation and processing. We thus propose the three ribotypes, supplemented with genome size estimation, as the key determinant for interspecific determination, as the thecal variability attributed to sample processing likely surpass that of nascent variation.

## 3. Discussion

Our data demonstrated significant inter-specific variations within the Crythecodiniaceae, with the ITS ribotype differentiating more cladal groups (Figure 3A), whereas the LSU rDNA and SSU rDNA ribotypes exhibited less variability (Figure 3B,C); all three ribotypes concurred with specific ITS insertion-deletion to substantiate *Crypthecodinium croucheri* sp. nov. *Crypthecodinium rykuyu* sensu lato *Crythecodinium okinawan* sensu lato. The N_1_ (the lowest haploid number) genome sizes with individual *C. cohnii* strains, despite minor differences in the thecal formula, were significantly and consistently higher than that of *C. croucheri* (~7.5–12%) (Figure 5A,B). This study points to the importance of N_1_ genome sizes, AFLP, and multi-ribotypes in species delineation of Crypthecodiniaceae. 

*C. croucheri* cells, which were significantly smaller than *C. cohnii* (in length: *p* = 0.0077, in width: *p* = 0.0231, unpaired *t*-test, Table 1), did not propagate well in the *C. cohnii* MLH medium or on MLH agar. Compared to *C. cohnii* strains that can grow across a wide range of temperatures (18–32 °C, Appendix A), *C. courcheri* exhibited a well-defined optimum thermal range (Appendix A) better suited to lower energy biotechnology, with less likelihood as invasive species. With the significantly higher cellular polar lipid (DHA) and total lipid content, further optimizations of culturing conditions and culture medium for *C. croucheri* will provide the basis for commercial production of DHA. 

Hong Kong is in subtropical zone with high precipitation in summer, less influenced from the Pearl River during winter, but affected by multiple oceanic water masses from the South China Sea [69]. These diverse ecosystems are associated with coastal algal blooms, very often caused by dinoflagellates [70], as well as having a repertoire of indigenous dinoflagellate species [71]. Crypthecodiniaceae members are commonly found along shorelines, likely with extremes of environmental conditions; bearing in mind, sexual reproduction was induced with 15–18 °C [30,33]. *C. cohnii*’s location in subtropical waters was proposed as a signal of warming in these waters [72]. *C. cohnii* (ATCC30556) was isolated from Tolo harbour in the northern part of Hong Kong, but its ribotype claded with other *C. cohnii* species rather than with *C. croucheri*. This may reflect epiphytic-benthic niche differentiation [73] or water quality parameter changes, as Tolo Harbor had become substantially freshwater with less seawater intrusion after the construction of Plover Clove Reservoir. The well-defined thermal range (Appendix A) also substantiates potential seasonal *C. croucheri* occurrences, as summer Hong Kong temperatures reach 33–35 °C.

It was proposed that global distribution of *C. cohnii* species were aided by tides and currents on which they attached on the surface of seaweed or other floating objects [8]. It is possible there are “epiphytic host” specificities amongst the different *Crypthecodinium* species. Residual bleached coral habitats are increasingly converted to macrophyte-dominated environments [74]. 

Crypthecodiniaceae members are heterotrophs commonly associated with seaweed [12,13]. As with many epiphytic dinoflagellates, *C. cohnii* cells formed extracellular exudates [75] and many cells could not be ‘washed away’, indicating potential forming of a phylloplane-like epiphytic community. It was proposed that the dinoflagellate carbon fixation was substantially underestimated attributed to the exudates [76], that we also found in Crypthecodiniaceae [75], which also provide substrate for microbial communities with implications in particulate organic carbon [77] and nutrient recycling.

The 0.04 genetic distance for inter-specific demarcation [42] was far exceeded for *C. cohnii-C. croucheri* and *C. cohnii-C. okinawan* and between *C. okinawan-C. croucheri*, but at lesser value for *C. cohnii-C. rykuyu* (Figure 1). The genetic distance between the proposed Crypthecodinales and the nearest Gonyaulacales (if rooting to *Amphidinium* e.g., *Alexandrium, Lingulodinium*) was ~1 to 1.2, far exceeding the genetic distance between the Gonyaulacales and the Peridiniales (e.g., *Heterocapsa, Cryptoperidiniopsoid*, ~1.0). The recent rectification of Symbiodiniaceae was mostly based on LSUrRNA ribotype [78] with smaller genetic distance to the nearest Order. The original placement of *C. cohnii* in *Peridinium* [6] was later changed to no less than five different orders including *Gyrodinium fucorum* Kofoid & Swezy, 1921 (unaccepted synonym), *Gyrodinium cohnii* Schiller, 1933 (unaccepted synonym) *Gymnodinium fucorum* Kuster, 1908 (unaccepted synonym), *Glenodinium cohnii* Seligo, 1887 (unaccepted synonym) and *Crypthecodinium setense* Biecheler, 1938 (unaccepted synonym) (World Registrar of Marine Species). *Glenodinium cohnii* was first described by Seligo (1885) [79], prior to transfer to *Gyrodinium cohnii* by Schiller (1933) [80]. Biecheler (1952) [68] described *Crypthecodinium setense* as the only species of the new genus, but while noting the similarity with *Gyrodinium cohnii* Schiller 1933. The grouping was considered co-specific and combined to *Crypthecodinium cohnii* (Seligo) Chatton in Chatton (1952) [67]. *C. cohnii* sensu lato was thus placed in *Peridinium* [3], *Gyrodinium, Gymnodinium, Glenodinium* and *Crypthecodinium.* The Crypthecodinale sensu lato node was more significant with inclusion of the R6 clade suggesting *bona fide* sub-order. The Crypthecodinale sensu lato long branch suggested substantial evolution from the nearest group, and comparison between Amphidiniaceae rooting, as reported with Ryukyu isolate analysis, supported the order’s being evolved from this early lineage.

The ITS differentiation harbored more hidden differentiation with truncation and insertion, amid consistency with the different strains isolated for both *C. croucheri* and *C. okinawan*. The vast majority of the insertions/truncations were conserved, with a small percentage of additional changes (e.g., ~5–7 and ~40 extra bases in the ITS1 and ITS2 region of *C. croucerhi*, respectively (Figure 2)), indicating that these highly recognized variable regions were unlikely to be neutral, which would have resulted in unequal transient insertion-deletion.

Oxidative stresses, which are abundant in intertidal zones, are well-appreciated factors in ribosomal biogenesis [81]. The stable high DHA content found in Crypthecodinale preparations is indicative of antioxidant effect, as revealed for a Crypthecodiniaceae [82]. *C. cohnii* was originally developed for carotenoid production [22], and we are comparing their interspecific levels for potential added value in biomass preparations.

Dinoflagellates are major producers of the anti-greenhouse gas dimethylsulfide (DMS) and related sulphur compounds, and *Crypthecodinium* spp. has one of the highest concentrations on record [83,84]. The high dinoflagellate DMSP content [84,85], which exhibited extracellular high antioxidant potential [86] in addition to the carotenoids [23] could have contributed to the stable DHA content after extraction; these major greenhouse-positive compounds have rendered this oleaginous group forefront candidates for next-generation sustainable biofuel [87]. *C. cohnii* and its biomass are well recognized as aquaculture feeds at different trophic levels [14,88,89]. In our search for sustainable aquaculture [90], the replacement of fish oil and fish meal are major milestones and heterotrophic conversion being a microbial looping platform for anthropogenic carbon, the potential conversion of carbon negative fixation, including those from macroalgae and domestic waste, will be twining hierarchical multitrophic environmental sustainability. Given the relatively small genome sizes [91], high cell density with defined medium and our recently developed transformation system [36,37], taxons of the Crypthecodiniaceae, and likely of Crypthecodiniales sensu lato, are promising platforms for biotechnology and synthetic biology, with the added benefit of high DHA content as active pharmaceutical ingredient (API).

## 4. Materials and Methods

### 4.1. Establishment of Clonal Cultures

Epiphytes were collected during low tides at Port Shelter on eastern Hong Kong (Table 1). To remove non-epiphytic species, surfaces of macrophyte and littoral samples were briefly rinsed three times with sterile f/2 medium before being placed in half strength f/2 medium at 24 °C without light [92]. Selected swarmer cells were purified by serial dilution in f/2 medium in 96-well plate. Potential strain contamination was checked with plating cells on 1:1 (*v*/*v*) f/2:rich medium (1% *w*/*v*) agar. This is important as purported bacterial stimulation of dinoflagellate bioactive compound, including DHA, would be difficult to reproduce.

All Crypthecodinale strains were maintained in either the MLH medium (for *C. cohnii*) [34] or half strength-Rich medium (16.7 g L^−1^ sea salts, 6 g L^−1^ glucose, and 2 g L^−1^ yeast extract, 1:1 (*v*/*v*) deionized water:sea water) [33] at 24 °C without light. *C. cohnii* Biecheler strain 1649 (CC1) was obtained from the Culture Collection of Algae at the University of Texas at Austin. *C. cohnii* (Seligo) Javornicky ATCC30021 (United States; Puerto Rico) and ATCC30556 (Shoreline organic debris, Tolo Harbor, Hong Kong, 1976) were obtained from American Type Culture Collection (ATCC, USA). 

### 4.2. Fluorescence Microscopy and Thecal Plate Formulation

Microscopic examination of the thecal plate tabulation was performed on glutaraldehyde-fixed (1% *w*/*v*) cells stained with calcofluor white (Fluka Analytical) [93]. The Kofoidian system was used for the designation of the thecal plate formula [26,66]. Light photomicrographs of cells were taken using the real-time extended depth of focus options with Nikon DS-Qi2 digital monochrome camera under ultraviolet excitation (340–380 nm).

Observation of cells was carried out with a stereomicroscope (Meiji EMT, Meiji Techno Co., Ltd., Saitama, Japan) and a binocular microscope (Leica DMLS) equipped with epifluorescence optics. The shape and position of nucleus was determined after staining of glutaraldehyde-fixed (1% *w*/*v*) cells with SYTO13 (0.025 mM, ThermoFisher Scientific, Waltham, MA, USA) for 5 min. Cell length and width were measured at 400× magnification using Lumenera Infinity Analyze software (Lumenera Infinity 3, Lumenera Corporation, Ottawa, ON, Canada) from photomicrographs. 

### 4.3. DNA Extraction, Amplification Fragments Length Polymorphism (AFLP), Genetic Distances and Phylogenetic Analyses

Genomic DNA extraction, PCR conditions, cloning and sequencing were conducted as described previously [94,95,96]. Sequence data of the Hong Kong isolates and other dinoflagellates’ rDNA genes (obtained from GenBank database) were aligned using the multiple alignment tool “ClustalW” in the MEGA7 software package (Pairwise and multiple alignment parameters—Gap opening penalty 10, Gap extension penalty 0.1, delay divergent sequences 30% and no use of a negative matrix) [97]. Phylogenetic trees were constructed based on the alignments by the maximum-likelihood method using Tamura-Nei (TN93) model in MEGA7 (Test of phylogeny options: Bootstrap 1000 replicates; Rates among sites: Uniform rates; Gaps/Missing Data Treatment: Complete Deletion). The resulting bootstrap values for each branch point are shown at nodes. Genetic distances (simple uncorrected pair-wise (p) distance) between ITS rDNA region (ITS1-5.8S-ITS2) sequences were calculated using MEGA7 and expressed as the number of substitutions per site. 

MrBayes version 3.2.7a was used to perform Bayesian analyses on ITS, SSU and LSU rDNA datasets [98]. The evolutionary model used in Bayesian analyses for all rDNA dataset was the GTR model with gamma-distributed rate variation across sites and a proportion of invariable sites. The following settings were used: four simultaneous Markov chain Monte Carlo (MCMC) run for 10 million generations, sampling every 100 generations. The first 25% of trees were discarded as burn in, and the remaining samples were used to infer Bayesian posterior probability at the nodes.

AFLP were conducted with PCR using total DNA preparations as templates, and primers for Cc18 and Cc20 (Appendix A) were based on sequences of *C. cohnii* repetitive elements [47]. Other non-*C. cohnii* taxons turned out to have only weak Cc20 PCR fragments when compared to *C. cohnii*. The Cc18 primers gave consistent strain-specific AFLP patterns, although sequencing of the PCR products suggested differences from the original reported sequence in *C. cohnii* (strains). Cc20 did not give enough resolution, as *C. croucheri* produced smearing pattern on the gel. It is likely the two markers targeting different chromosomal compartments (data not shown).

### 4.4. Flow Cytometric Analyses of Cellular Lipid Content and Nuclear Genome Size

Nile red (Sigma-Aldrich, St. Louis, MO, USA, N3013; final concentration: 0.1 µg mL^–1^) staining was performed, as described previously [53]. Nile-red-stained cells were examined with yellow-gold fluorescence (excitation, 488 nm; emission, 530 ± 30 nm) for neutral lipids and red fluorescence (excitation, 488 nm; emission, 575 ± 26 nm) for polar lipids [53,54,99]. Nile-red-stained cells were also co-stained with DAPI (2.5 µg mL^−1^), and analyzed by a Becton Dickinson FACSAria™ IIIu cell sorter (BD Biosciences) for estimating the cellular lipid content in different cell cycle phases. Nile-red-stained *C. croucheri* cells consistently gave higher oil content than *C. cohnii* cells (CC1, ATCC30021 and ATCC30556) when grown in comparative conditions.

Comparison of nuclear genome sizes was made with standard model organisms with published genome sizes, including cells of the budding *yeast Saccharomyces cerevisiae* (BY23849, National BioResource Project, Japan), and cultured mammalian cells (*HL-60, ATCC*). HL-60 cell line has a diploid genome size of 6.5 pg per cell [48], whereas the haploid genome sizes of *S. cerevisiae* and *Symbiodinium minutum* is 0.017 and 1.5 pg per cell, respectively [49,50]. Propidium iodide (PI) staining was performed as described previously [100]. PI-stained cells were analyzed by a Becton Dickinson FACSAria™ IIIu cell sorter (BD Biosciences) and flow cytograms were computed with the software FlowJo (version X 10.0; The Tree Star, Inc., Ashland, OR, USA). To facilitate direct comparison, values of G_1_ cells were selected from flow cytograms for comparison, using the “Range” function available in the FlowJo software. For more stringent comparison of genomes’ sizes, we deployed only G_1_ peak (haploid genome size) as *C. cohnii* cells likely composed of cells going through different partial decompaction with S phase [101], and that multiple fission cells [28] would have increased the G_2_ peak width.

### 4.5. Biochemical Quantification of Lipid Content

Lipid contents of *C. cohnii* and *C. croucheri* were also measured by the Sulpho-phospho-vanillin (SPV) method [62,63,64]. The Phospho-vanillin reagent was prepared freshly by dissolving 0.075 g vanillin in 12.5 mL distilled water and mixed with 50 mL 85% phosphoric acid. Cells were counted, harvested with low-speed centrifugation (1000× *g*, 10 min) and washed two more times with distilled water. The pellets were freeze-dried and dry weights (until constant weight) were calculated by subtracting the weight of the empty tubes. Dried pellets were resuspended in 1 mL of chloroform: methanol (at 2:1 (*v*/*v*)) and sonicated (Amplitude = 70%, Duration = 30 s, 3 s on-3 s off cycle) on ice, followed by centrifugation at 16,000× *g* for 15 min. This extraction process was repeated three times. The three supernatants were collected (in glass vial) and dried at 90 °C. The standard curve was prepared by using different amounts (0–150 µg) of commercial canola oil (obtained from local market) as a standard. Concentrated sulphuric acid (0.1 mL) was added to each vial, and the suspension was boiled for 10 min and cooled on ice for 5 min. Phospho-vanillin reagent (2.4 mL) was added and allowed to develop for 15 min until the color of the samples turned pink. Absorbance at 492 nm was measured using the Multiskan FC Microplate Photometer (Thermo Scientific).

### 4.6. Statistical Analysis 

Experiments were conducted in triplicate, repeated three times with similar results, and data were presented as mean value ± standard error (SE). Unpaired *t*-test statistical analysis was performed using GraphPad Prism. Results were considered significant when *p* < 0.05.

## Figures and Tables

**Figure 1 marinedrugs-21-00162-f001:**
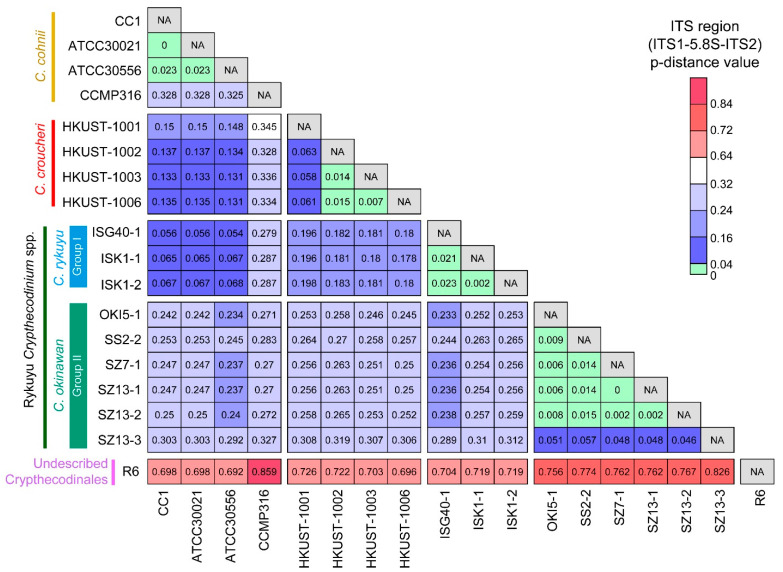
Uncorrected nucleotide genetic distances (*p*-distances) of the ITS region between Crypthecodiniaceae and other dinoflagellates. Values lower than 0.04, which was suggested as the species-level divergences cut-off, are highlighted in green color [42].

**Figure 2 marinedrugs-21-00162-f002:**
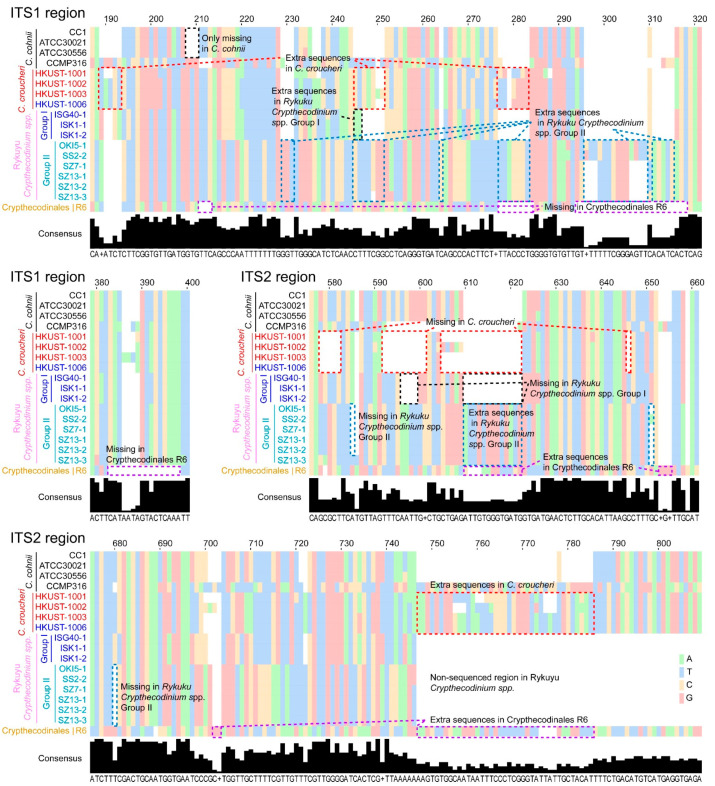
ITS regions exhibited intra-specific truncations-insertions within the Crypthecodiniaceae. This schematic diagram shows the location of extra and missing nucleotide stretches observed in the ITS1 and ITS2 regions in *C. cohnii*, *C. croucheri* and Ryukyu *Crypthecodinium* spp. See Appendix A for the full set of ITS1-5.8S-ITS2 sequence alignments.

**Figure 3 marinedrugs-21-00162-f003:**
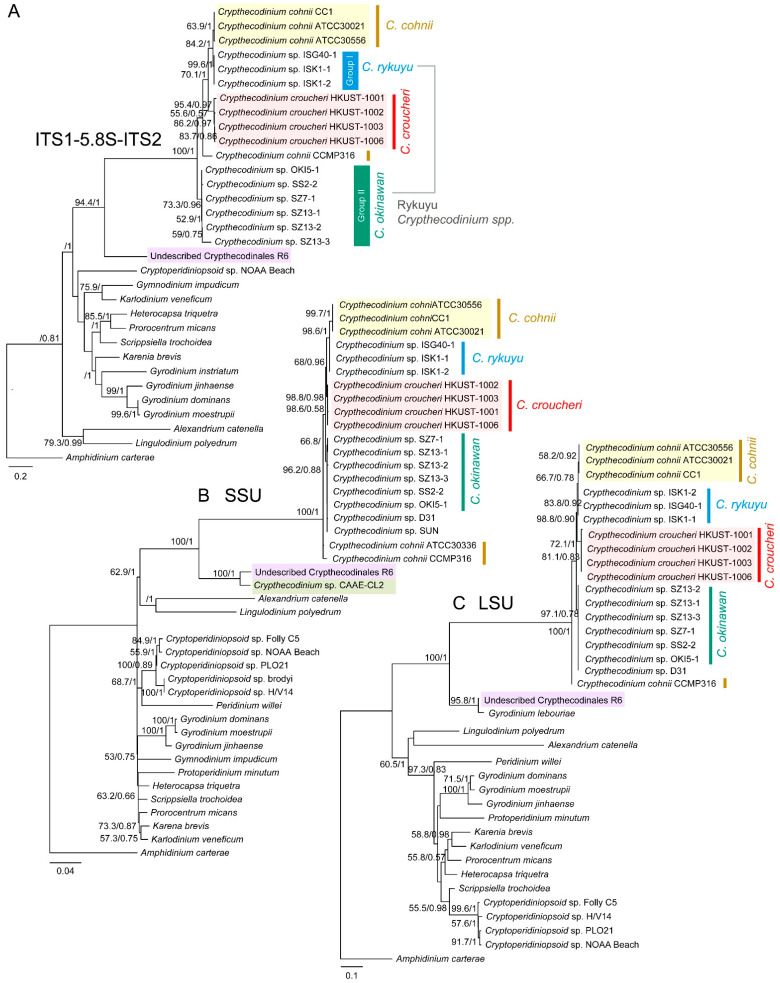
Rooted phylogenetic trees constructed with ribosomal DNA sequences of some Crypthecodiniaceae members. Bayesian and maximum likelihood (ML) combined phylogenetic (rooted) trees of Crypthecodiniaceae derived from (**A**) ITS rDNA, (**B**) SSU rDNA and (**C**) LSU rDNA sequences. Numbers at nodes represent the bootstrap support values (BP, in %, based on 1000 replicates)/posterior probabilities (PP) produced by ML and Bayesian inference analyses respectively. Values > 50% for BP and >0.5 for PP are shown. The ML and Bayesian tree topologies were congruent with each other. All trees were rooted with *Amphidinium carterae*, a recognized early lineage.

**Figure 4 marinedrugs-21-00162-f004:**
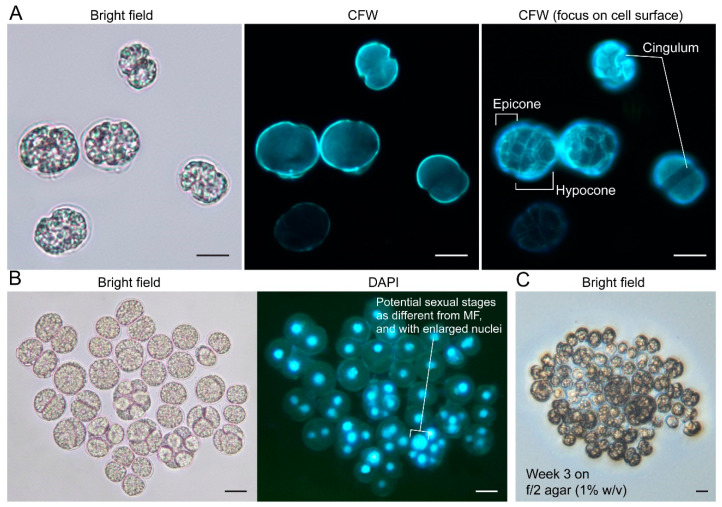
*Crypthecodinium croucheri.* sp. nov.: (**A**) fluorescence photomicrographs of calcolfuor white (CFW)-stained cells; (**B**) Light and fluorescent (DAPI) photomicrographs showing examples of multiple fissions, also served to show lack of bacterial contamination and (**C**) formed colonies on 1% (*w*/*v*) f/2-agar plate. Scale bar = 10 µm.

**Figure 5 marinedrugs-21-00162-f005:**
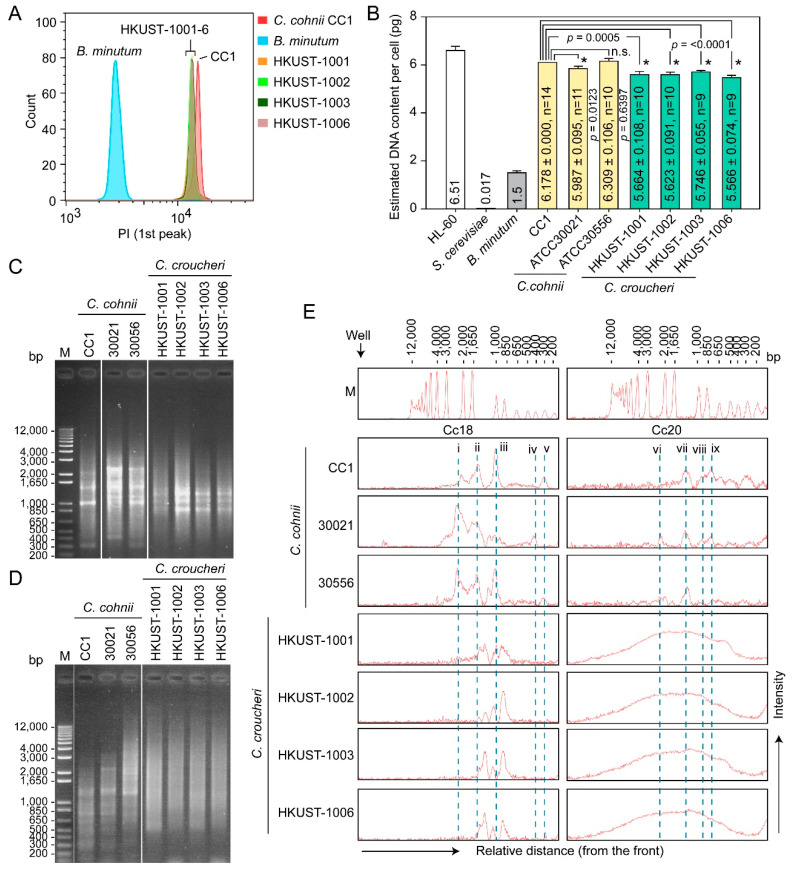
Comparative genome sizes and AFLP patterns of Crypthecodiniaceae strains: (**A**) Flow cytogram overlays of propidium iodide (PI)-stained *Crypthecodinium cohnii* CC1, *C. croucheri* sp. nov (HKUST-1001, 1002, 1003, 1006), and *Breviolum minutum*. Only G_1_ DNA-peak (haploid) was shown. (**B**) By including samples with known genome size (HL-60—6.51 pg cell^−1^ [48], *B. minutum*—1.5 pg cell^−1^ [49], *Saccharomyces cerevisiae*—0.017 pg cell^−1^ [50]) (R^2^=0.9983), the relative (mode) PI fluorescence was used to calculate the genome size of the *C. croucheri* and *C. cohnii* samples. Asterisks (*) indicate significant difference from *C. cohnii* CC1 (unpaired *t*-test, *p* < 0.05). n.s. = no statistical difference. AFLP fingerprint patterns of Crypthecodiniaceae strains were generated using (**C**) Cc18 and (**D**) Cc20 primer pairs. M = 1 kb plus DNA ladder (Invitrogen). (**E**) Densitometric profile of the AFLP-generated DNA bands as shown in (**C**) and (**D**). Small Roman “i–v” and “vi–ix” correspond to Cc18 primers-and Cc20 primers-amplified DNA bands specific to *C. cohnii*, respectively. Densitometric analysis was performed using tools native to Image Lab software (Bio-Rad, version 6.1).

**Figure 6 marinedrugs-21-00162-f006:**
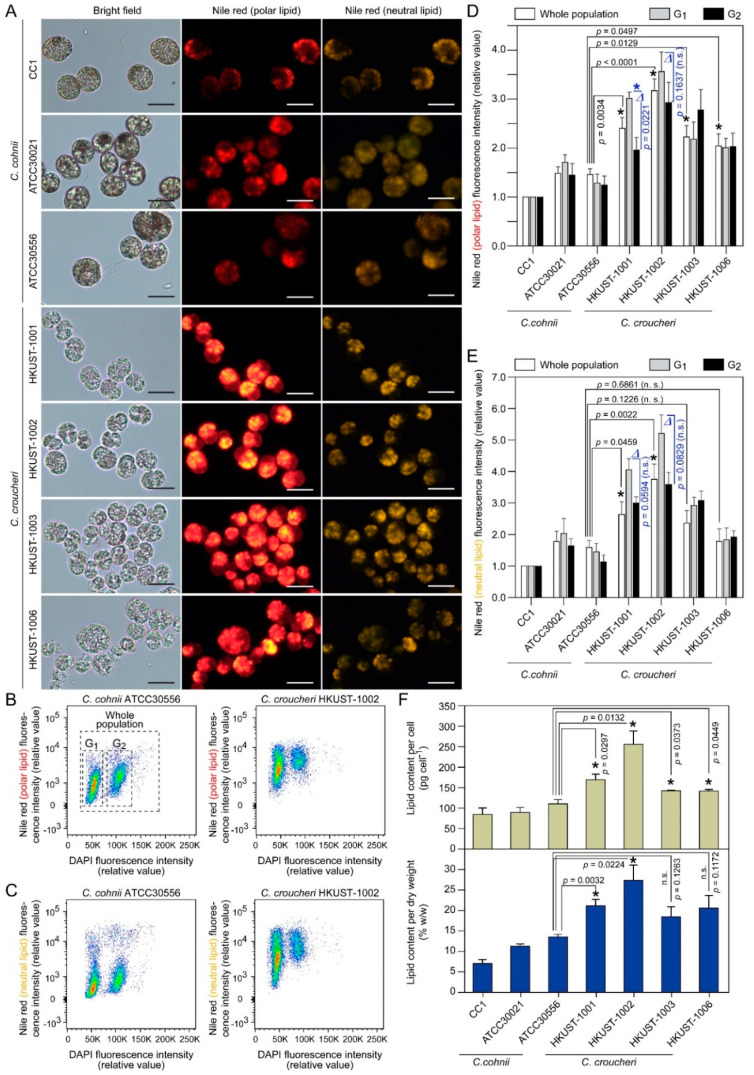
Lipid content of *Crypthecodinium* cells: (**A**) Light and fluorescence photomicrographs of Nile-red-stained cells. Nile-red-stained cells were imaged at the corresponding emission spectrum for polar lipids (red fluorescence) and apolar (neutral) lipids (golden yellow fluorescence). Scale bar = 10 µm. (**B**,**C**) Flow cytograms of *C. cohnii* ATCC30556 and *C. croucheri* HKUST-1002 cells co-stained with DAPI (for relative DNA content) and Nile red for (**B**) polar and (**C**) neutral lipid content. Flow cytometric analysis of Nile-red-stained *C. croucheri* cells suggested higher (**D**) polar lipid and (**E**) neutral lipid content per cell (either G_1_ or G_2_ cells), when compared to *C. cohnii.* The symbol “∆” marked the widened differences between G_1_ and G_2_ cells’ lipid content in *C. croucheri* HKUST-1001 and HKUST-1002, which were not observed in *C. cohnii*. As *Crypthecodinium* cellular oil content could differ vastly in different genome cycle phases, it is important to compare cell cycle gated values. (**F**) Total lipid content (per cell and per dry weight) of *Crypthecodinium* cells were independently quantified by using the sulpho-phospho-vanillin method, as described previously [62,63,64]. All cells were batch-grown at 24 °C in 1:1 (*v*/*v*) f/2:rich medium for 7 days without light. Data represent means ± SE of three replicate experiments. Asterisks (*) indicates significantly different from the *C. cohnii* ATCC30556 (*p* < 0.05). n.s. = non-significant (*p* > 0.05).

**Figure 7 marinedrugs-21-00162-f007:**
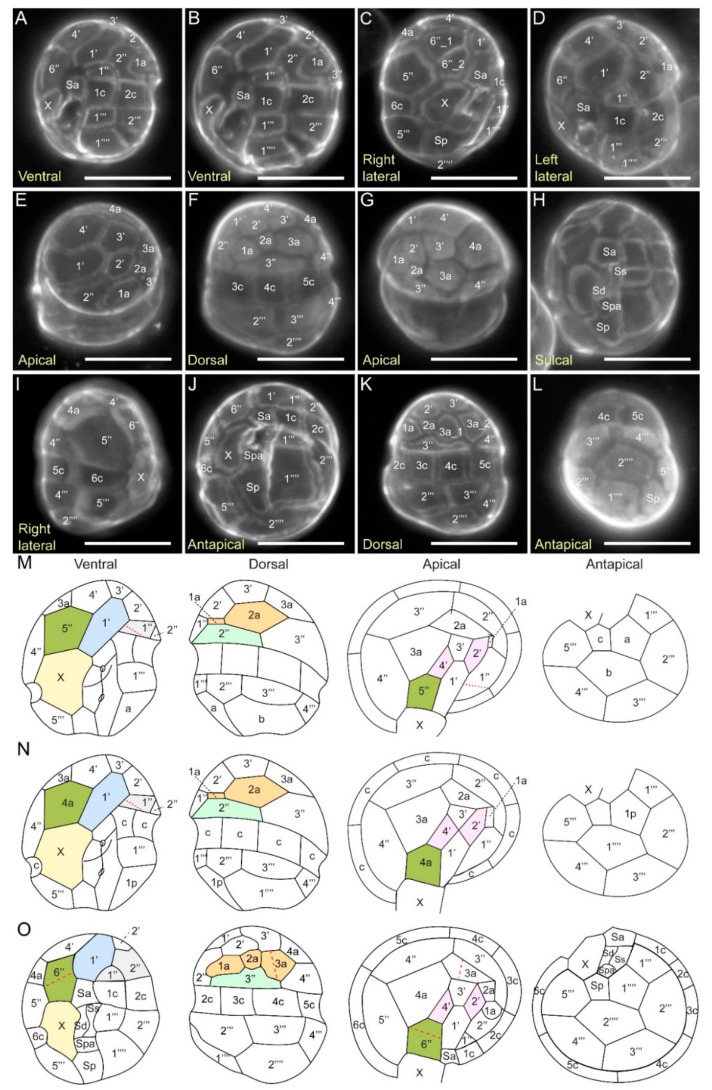
Thecal plate patterns of *Crypthecodinium croucheri.* sp. nov. and other Crypthecodiniaceae Calcolfuor white (CFW) fluorescence photomicrographs showing the thecal pattern of *Crypthecodinium croucheri* in ventral (**A**,**B**), dorsal (**F**,**K**), right lateral (**C**,**I**), left lateral (**D**), apical (**E**,**G**), antapical (**J**,**L**) and sulcal (**H**) views; Plate labelling based on Kofodian system. Sa = anterior sulcal plate; Ss = left sulcal plate; Sd = right sulcal plate; Spa = posterior accessory plate; Sp = posterior sulcal plate. Scale bar = 10 µm. Line drawing of the thecal plate patterns of (**M**) *Crypthecodinium setense* [6], which was later synonym with *Crypthecodinium cohnii (Seligo) Chatton* [67] and *Gyrodinium cohnii* (Seligo) Schiller [68], (**N**) *Crypthecodinium* spp. CAAE-CL2, previously isolated form brackish-water fish aquarium [9]. and (**O**) *Crypthecodinium croucheri.* Redrawn with permission from [9]. Copyright © 2006, Taylor & Francis (License number: 5480661155043). Variation between *C. cohnii* and *C. croucheri*: the *C. croucheri* precingular plate 6″ splits into two plates, 6″1 & 6″_2 (**C**,**O**) and the anterior intercalary plate 3a splits into 3a_1 & 3a_2 (**K**,**O**). Sa = anterior sulcal plate; Ss = left sulcal plate; Sd = right sulcal plate; Spa = posterior accessory plate; Sp = posterior sulcal plate.

**Table 1 marinedrugs-21-00162-t001:** Cell sizes of *Crypthecodinium croucheri*, sp. nov., *Crypthecodinium cohnii*, *Crypthecodinium rykuyu* sensu lato, and *Crypthecodinium okinawan* sensu lato.

		*C. cohnii* ATCC30556 (*n* = 24)	*C. croucheri* HKUST-1002 (*n* = 31)	*C. rykuyu* (Rykuyu Isolate Group I) (*n* = 3) *	*C. okinawan* (Rykuyu Isolate Group II) (*n* = 6) *
Length (μm)	Mean	16.9 ± 2.1	15.0 ± 2.8	10.3 ± 0.7	9.9 ± 0.6
	Minimum	13.9	10.7	9.5	8.9
	Maximum	21.3	21.1	11.0	10.4
Width (μm)	Mean	14.1 ± 2.0	12.6 ± 2.6	8.7 ± 0.0	8.4 ± 0.6
	Minimum	11.2	7.9	8.7	7.4
	Maximum	18.2	18.6	8.8	9.0
Mean length/width ratio		1.20	1.22	1.19	1.18

***** Cell size was estimated directly from Figure 1 of Prabowo et al., 2013 [12]. The authors claimed the cell size ranged from 5–30 μm.

## Data Availability

The data presented in this study are openly available in the GenBank database and their accession numbers are listed in Appendix A.

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
