# Peer review of "Oleaginous Heterotrophic Dinoflagellates—Crypthecodiniaceae"

_marinedrugs, 2023, doi:10.3390/md21030162_

Round 1

Reviewer 1 Report

Kwok et al have done a remarkable analysis of Crypthecodinium croucheri by combining different approaches. The paper is well written with minor need for rephrasing. As for the phylogenetics, I encourage the addition of Bayesian inference to the current maximum likelihood method so as to provide more support to the current position within the tree. Below are minor corrections. 

Introduction

Line 27: The word " indicated" does not seem to fit here. I suggest authors to change it.

Line 39-40: Please rephrase this sentence as it is in past tense and no reference provided.

Line 41: It is unclear what the authors mean by "was indicated". Pease clarify.

Line 49-50: It feel like something is missing in this sentence "colony-forming capacity, true minimal medium without undefined seawater"

Line 54: Again something is missing here "the group also for biofuel production"

Results:

Line 91-92: It is not clear what the authors mean by "were harbored" in this sentence.

Line 109: Maybe the authors mean substantial. Please confirm.

Line 111: trancations -> truncations

line 112: alignemnt -> alignments

line 115: dinflagellate -> dinoflagellates

Line 183: C. roucheri cells. ->  croucheri

Line 314: the the. Duplication of "the".

Discussion:

Line 417: linages -> lineages

Table 2 & 3 can be provided as supplementary materials. 

Methods:

Line 518: Since the major conclusion of this study relies with the use of a single phylogenetic method (maximum-likelihood method) , i suggest the possibility to include another approach such as bayesian inference and provide those values in the phylogenetic tree.

Author Response

We thank the editor and the reviewers for their prompt attention and suggestions.

We include additional data and literature, as well as addition phylogenetic analysis per the suggestions of the reviewers.

We hope the manuscript is acceptable for publication in Marine Drugs.

Please see the attachment for the point-by-point response to the reviewer’s comments.

Reviewer 2 Report

Comment to the paper: “Oleaginous heterotrophic dinoflagellates - Crypthecodiniaceae” by Alvin Chun Man Kwok et al.

 General comment:

The manuscript deals with investigations on genetic distances and phylogenetic cladding that support inter-specific variation within the Crypthecodiniaceae.

The manuscript is suitable to be published in this journal, however some points should be addressed before publication.

Some minor language mistakes are present that should anyway be corrected.

1. Introduction

Please, improve the literature overview on value-added compound extraction from microalgae. Please, consider the following papers:

o   Enhancing Biomass and Lutein Production From Scenedesmus almeriensis: Effect of Carbon Dioxide Concentration and Culture Medium Reuse (2020) Frontiers in Plant Science, 11, art. no. 415.

o   New insights into the stoichiometric regulation of carotenoid production in Chlorella vulgaris (2022) Bioresource Technology Reports, 20, art. no. 101227

o   Lutein production with Chlorella sorokiniana MB-1-M12 using novel two-stage cultivation strategies – metabolic analysis and process improvement (2021) Bioresource Technology, 334, art. no. 125200

4. Materials and methods

Please, clarify if the characterization of value added compounds was carried out.

Please, specify if the variation of the growth condition was carried out.

2. Results

Please, include lipid characterization.

3. Discussion

Please, improve comparison between your findings and literature data.

Author Response

(The authors gave the same response as above.)
